# A Service Life Prediction Method of Stranded Carbon Fiber Composite Core Conductor for Overhead Transmission Lines

**DOI:** 10.3390/polym14204431

**Published:** 2022-10-20

**Authors:** Yongli Liao, Ruihai Li, Chuying Shen, Bo Gong, Fanghui Yin, Liming Wang

**Affiliations:** 1Electric Power Research Institute, China Southern Power Grid Co., Ltd., Guangzhou 510663, China; 2Shenzhen International Graduate School, Tsinghua University, Shenzhen 518055, China

**Keywords:** carbon fiber composites, thermogravimetric analysis, activation energy, reaction mechanism function, life prediction

## Abstract

The effect of temperature on the service life of stranded carbon fiber composite core conductors was studied based on the kinetic theory of material pyrolysis. The thermal decomposition activation energy calculation for stranded carbon fiber composite cores was carried out by thermogravimetric analysis (TGA). The activation energy *E* of stranded carbon fiber composites was calculated according to the Flynn–Wall–Ozawa, Kissinger, and Coast–Redfern methods, which were 168.76 kJ/mol, 166.79 kJ/mol, and 160.35 kJ/mol, respectively. The results from these different treatments were within 10% or less, and thus the thermochemical reactions of stranded carbon fiber composite cores were considered to be effective. The life prediction model of the carbon fiber composite core was developed based on the kinetic equation of thermal decomposition. The service life is related to the reaction mechanism function G(*α*) and the reaction rate parameter k(*t*). The reaction mechanism function G(*α*) = ((1 − *α*)^−3.3^ − 1)/3.3 and the reaction rate parameter k(*t*) = 2.14 × 10^12^exp(*E*/*RT*) were obtained by fitting the thermal weight loss data of stranded carbon fiber composite cores. Based on the 5% mass loss criterion for the end of life of stranded carbon fiber composites, the service life of the carbon fiber composite core is given at various operating temperatures.

## 1. Introduction

The inner core of a stranded carbon fiber composite core conductor is composed of multiple strands (typically seven) of carbon fiber composite cores [1,2], as shown in Figure 1. The single-stranded carbon fiber composite core consists of a composite of carbon fiber filaments and epoxy resin with an outer protective glass fiber layer. The aluminum conductor is generally annealed soft aluminum or heat-resistant aluminum alloy. Compared with steel-core aluminum stranded conductors of the same diameter, the carbon fiber composite core conductor has smaller mass, lesser thermal expansion and arc sag, and is resistant to high temperatures of 160–180 °C [3,4,5,6,7]. This type of conductor is widely used in capacity increase and renovation projects on old lines. The original transmission line’s conventional conductor is replaced with the carbon fiber composite core conductor (a high temperature-resistant conductor). Without changing the original line path and without replacing the tower, the maximum arc sag to ground after replacing the carbon fiber composite core conductor is still less than the maximum arc sag to ground of the steel-core aluminum stranded conductor (original conductor), which has an economic advantage in transforming old lines [8,9,10,11].

The design life of transmission line equipment is generally 30–50 years [12,13]. The service life of stranded carbon fiber composite core conductors is mainly determined by the carbon fiber composites. Carbon fiber is an inorganic polymer fiber material containing more than 90% carbon, with excellent heat resistance, and does not decompose under high-temperature conditions [14,15,16]. Epoxy resins decompose under environmental conditions such as high temperature. Experimental results have shown that under the action of the external environment, the resin matrix of carbon fiber composites will undergo certain kinds of damage, leading to the degradation of the bending properties of carbon fiber composite cores [17]. The physical mechanisms and mechanisms of the occurrence and development of thermal aging of carbon fiber composites are not fully understood, so it is difficult to predict the service life of a carbon fiber composite core conductor. When power grid operation departments use carbon fiber composite core conductors, they will not be able to fully grasp whether the operational status of a carbon fiber composite core conductor is safe or not.

The overhead transmission line conductor service environment is more complex. The atmospheric environment is susceptible to pollution, moisture, and a variety of other factors that lead to corrosion degradation of conductor performance [18,19,20,21,22]. Different material contact surfaces are prone to galvanic corrosion problems. The galvanic corrosion behaviors between carbon fiber composites and aluminum have been reported [23,24,25]. In order to compare the corrosion characteristics of stranded carbon fiber composite core conductors and galvanized steel–aluminum conductors, a 2000 h salt spray test was conducted. A stranded carbon fiber composite core conductor has a protective layer surrounding the composite core, which makes the galvanic corrosion less severe compared to the steel-core aluminum stranded conductor [26].

Although the stranded carbon fiber composite core has a protective layer outside with strong corrosion resistance, aging of the carbon fiber composite is always a problem. Many scholars have used various methods to study the service life of composite materials for power facilities [27,28,29]. Application of Fickian theory for glass fiber-reinforced epoxy matrix composites permits predicting the effect of humidity on composite lifetime. A linear fit of the conductor performance versus time was performed according to the Arrhenius model for a degradation life prediction of polymer matrix composites. The least squares method was used to obtain the degradation equation of the material performance parameters. The thermal aging equation was used to achieve the failure year of carbon fiber composite core conductors, with a 30% decay of the conductor performance as a criterion. The creep behavior of the carbon fiber composite core conductor was carried out at different temperatures and mechanical stresses. The creep performance at high temperature and high mechanical stresses for a short period of time was used to estimate the creep at low temperature and low mechanical stresses for a long period of time. This accelerated test method can be used to predict the service life of the conductor.

In recent years, some scholars have used the intrinsic properties of materials (activation energy) to study the characterization of their deterioration processes [30]. The activation energy, which is an intrinsic property of the material, helps to reveal the mechanism of the onset and development of material deterioration. The activation energy parameters were used to assess the remaining life of insulated cables and insulators, and the service life was calculated using thermal aging experiments and state parameters [31,32,33,34]. The chemical reaction rates and activation energies of materials at different temperatures can be evaluated using the TGA method [35]. For the aging of carbon fiber composites, it can be considered that the main cause is thermal stress. The deterioration process of the material and its operational life depend mainly on the thermal effect. Under the action of thermal stress, the material will continuously undergo aging with time, and the degradation of the mechanical properties of the carbon fiber composite core will occur.

In this paper, stranded carbon fiber composite core samples were taken as an example, and the heat loss curve variation of carbon fiber composite samples was obtained by conducting heat loss tests at different heating rates. By calculating the activation energy, reaction mechanism function, and pre-exponential factor of carbon fiber composites, the remaining life of carbon fiber composites at different temperatures was evaluated, and the influencing factors are discussed.

## 2. Pyrolysis Kinetics Model

The reaction kinetic equation followed by solid composites during pyrolysis has the following form [36]: (1)$$\frac{d\alpha }{dt}=k\left(T\right)f\left(\alpha \right),$$
where *α* is the reaction conversion rate, reflecting the degree of decomposition; k(*T*) is the reaction rate constant, which is a function of temperature; and f(*α*) is the reaction mechanism function.

The reaction conversion rate *α* refers to the ratio of the decomposed part to the total weight lost at a certain time, and the formula for the conversion rate can be obtained according to the definition as follows: (2)$$\alpha =\frac{m_{i}-m}{m_{i}-m_{f}},$$
where *m* is the mass at a particular time and temperature, *m_i_* is the initial mass, and *m_f_* is the final mass [37].

The reaction rate constant k(*T*) is the rate constant–temperature relationship described by the Arrhenius equation, expressed as follows: (3)$$k\left(T\right)=A\exp\left(-\frac{E}{RT}\right),$$
where *A* is the pre-exponential factor, *E* is the activation energy for the decomposition reaction, *R* is the gas constant (8.314 J/mol·K), and *T* is the thermodynamic temperature.

By substituting Equation (3) into Equation (1) and dividing both sides by the heating rate *β*(d*T*/d*t*) at the same time, the kinetic equation of thermal decomposition can be obtained as follows:(4)$$\frac{d\alpha }{dT}=\frac{A}{\beta }\exp\left(-\frac{E}{RT}\right)f\left(\alpha \right),$$

Subsequently, the three kinetic state parameters of pyrolysis, namely activation energy, reaction mechanism function, and the pre-exponential factor, can be derived from the thermal weight-loss curve based on Equation (4).

## 3. Pyrolysis Kinetic State Parameters

### 3.1. TGA Test

The carbon fiber composite core rod applied to overhead conductors was taken as the test object. The protective capping layer of the carbon fiber composite core rod composed of organic fiber wires was filed off. After wiping the surface of the carbon/epoxy composite material sample with alcohol, it was ground into powder and placed in a drying oven at 40 °C for drying. 

The TGA test was carried out using a TGA 4000 (PerkinElmer, Waltham, Massachusetts, USA) instrument. The test was carried out in a dry flow rate of 100 mL/min nitrogen atmosphere. Samples of 5–10 mg were taken for each test, and the heating rates were set at 5, 10, 20, and 25 K/min. The temperature range of the test was from room temperature to 900 °C.

The thermogravimetric (TG) curves and the derivative thermogravimetry (DTG) curves of carbon fiber composite cores at different heating rates are shown in Figure 2. The DTG curve is the first-order differential curve of the TG curve, which represents the decomposition rate from thermogravimetric data. It can be seen that with an increase in the heating rate, peaks in both the TG and DTG curves move toward high temperature. The onset decomposition temperature, the termination decomposition temperature, and the temperature at the maximum decomposition rate (i.e., the peak temperature in the DTG curve) all increased slightly. The changes in the thermal weight loss of the samples at different heating rates showed the same trend, indicating that the reaction mechanism of the carbon fiber composite core was consistent at different heating rates. As can be seen from Figure 2b, when the temperature was greater than 600 °C, the DTG curve tended to zero, that is, the thermal decomposition ended. It can be seen from Figure 2a that the quality tended to be stable at this time. Carbon fiber is a kind of inorganic polymer fiber material with a carbon content higher than 90% and has excellent heat resistance [38]. After reaching a temperature of 600 °C, the epoxy resin material in the carbon fiber composite has been decomposed, and the remaining components are mainly carbon fiber and carbon residue.

### 3.2. Activation Energy

Various classical models have been proposed for calculating the activation energy of materials, such as the Flynn–Wall–Ozawa method based on the integral form [39] and the Kissinger method based on the differential form [40]. Combined with the results of the TGA test of carbon fiber composites carried out at different heating rates, the activation energy of carbon fiber composites can be obtained based on the above calculation models.

The Flynn–Wall–Ozawa method is an integral-based dynamic method with the following equation:(5)$$\ln\beta =\ln\left(\frac{AE}{RG\left(\alpha \right)}\right)-5.3305-1.0516\frac{E}{RT},$$
where the integral form of reaction mechanism function G(*α*) is as follows:(6)$$G(\alpha )=\int_{0}^{\alpha }\frac{d\alpha }{f(\alpha )},$$

Since the DTG curve peak temperature corresponds to each *α* approximately equally at different heating rates, ln(AE/RG(α)) can be considered as constant. The ln(*β*) in Formula (5) satisfies a linear relationship with 1/*T*, and the value of activation energy *E* can be obtained from the slope. The advantages of this method is that the activation energy can be solved without setting the reaction mechanism function; it avoids the possible calculation error caused by the different reaction mechanism functions, and it is often used to test the activation energy value by assuming the reaction mechanism function.

The activation energy of the carbon fiber composite core was calculated by the Flynn–Wall–Ozawa method. The peak temperature (*T**_pi_***) at different heating rates (*β*) read from the DTG curve (Figure 2b) is shown in Table 1. At the peak temperature of the DTG curve, the thermal decomposition rate of the carbon fiber composite was the largest. According to Formula (5), for ln(*β*) and 1/*T*, as shown in Figure 3, the activation energy is 168.76 kJ/mol.

The linear fitting curves and activation energy results at different conversion rates based on the Flynn–Wall–Ozawa method are shown in Figure 4. It was found that the activation energy values were basically stable within a certain range. In the thermogravimetric analysis (TGA) of carbon fiber-reinforced epoxy, the carbon fiber did not decompose. The thermal degradation of composite cores is predominantly due to the decomposition of the epoxy resin [21]. As the temperature increases, the physical and chemical cross-linked epoxy resin molecules began to break. More energy is required, and the activation energy reaches the maximum. The average activation energy was obtained as 175.24 kJ/mol by averaging the activation energies at different conversion rates. 

The Kissinger method proposes another way to calculate the activation energy, assuming that the reaction mechanism function is as follows: (7)$$f\left(\alpha \right)={\left(1-\alpha \right)}^{n},$$

Combining Equation (7) and Equation (4), and differentiating the two sides, the following results are obtained. The thermal decomposition reaction rate is the highest at the peak temperature *T_pi_* of the curve. From the TGA test under different heating rates, a set of corresponding *T_pi_* values can be obtained. Through linear fitting, the slope is equal to *-E/R*, and then the value of activation energy *E* can be obtained.
(8)$$\ln\left(\frac{\beta }{T_{pi}^{2}}\right)=\ln\left(\frac{AR}{E}\right)-\frac{E}{R}\frac{1}{T_{pi\;}},$$

The peak temperature (*T**_pi_***) of the DTG curve is the same as Table 1, and the fitting curve is shown in Figure 5. According to Formula (8), for ln(*β/T^2^*) and 1/*T*, as shown in Figure 5, the activation energy is 166.79 kJ/mol at the maximum thermal decomposition rate of peak temperature (*T**_pi_***). The calculated results are not much different from those obtained by the Flynn–Wall–Ozawa method, 168.76 kJ/mol.

### 3.3. Reaction Mechanism Function

The complexity of the thermal decomposition reaction can be judged from the conversion–activation energy relationship. According to the relationship between activation energy and conversion rate obtained earlier, it can be seen that the activation energy value varies very little with the conversion rate, and the reaction process can be considered to follow a single reaction mechanism function. The simultaneous integration of both ends of the pyrolysis kinetic Equation (4) yields the following equation:(9)$$G\left(\alpha \right)=\int_{T_{0}}^{T}\frac{A}{\beta }\exp\left(-\frac{E}{RT}\right)dT\approx \int_{0}^{T}\frac{A}{\beta }\exp\left(-\frac{E}{RT}\right)dT=\frac{AE}{\beta R}P\left(u\right)$$

Considering the low temperature at the beginning of the reaction *T*_0_, the reaction rate is negligible and both sides can be integrated between 0~*α* and 0~*T*. P(*u*) in the formula is called the temperature integral, and the expression is as follows, where *u* = *E/RT*:(10)$$P\left(u\right)=\int_{-\infty }^{T}-\frac{e^{-u}}{u^{2}}du,$$

Since P(*u*) has no analytical solution mathematically, an approximate solution can be obtained from the Doyle integral approximation formula [41]:(11)$$P\left(u\right)=0.00484e^{-1.0516u},$$

If *α* = 0.5 is assumed as the reference point [42], it can be obtained as follows:(12)$$G\left(0.5\right)=\frac{AE}{\beta R}P\left(u_{0.5}\right),$$

It is further obtained as follows:(13)$$\frac{P\left(u\right)}{P\left(u_{0.5}\right)}=\frac{G\left(\alpha \right)}{G\left(0.5\right)},$$

A series of experimental data points can be obtained by substituting the activation energy *E* and temperature *T* obtained at different conversion rates (Figure 4) into the left end of Equation (13). The experimental data points at different heating rates are given in Figure 4, and it can be seen that the data points at the four heating rates almost all overlap, indicating that the thermal weight loss process of the carbon fiber composite core follows a single reaction mechanism function. The trend of these data with the conversion rate coincides with the reaction mechanism function assumed in Equation (7), i.e., it follows f(*α*) = (1 − *α*)*^n^* and G(*α*) = ((1 − *α*)^−*n*^ − 1)/*n*. As shown in Figure 6, the best fit to the experimental data was found after several correction calculations when *n* = 3.3, and thus the reaction mechanism function of the carbon fiber composite core was determined as G(*α*) = ((1 − *α*)^−3.3^ − 1)/3.3.

### 3.4. Pre-Exponential Factor

The Coast–Redfern method [43] was used to incorporate the obtained reaction mechanism function, and the pre-exponential factor *A* was obtained from the curve fitting to obtain the following equation:(14)$$\ln\left[\frac{G\left(\alpha \right)}{T^{2}}\right]=\ln\left(\frac{AR}{\beta E}\right)-\frac{E}{RT},$$

In Equation (14), ln[G(*α*)/*T*^2^] and 1/*T* satisfy the linear relationship, the pre-exponential factor *A* is obtained from the intercept of the fitting curve, and the activation energy *E* is obtained from the slope of the fitting curve. The thermal weight loss test data were selected and combined with the obtained reaction mechanism function for fitting calculation, and the fitting curve was obtained as shown in Figure 7. From this calculation based on the Coast–Redfern method, the activation energy *E_C_* was 160.35 kJ/mol, and the finger front factor *A_C_* was 2.14 × 10^12^ s^−1^.

The activation energy *E_C_* and the finger front factor *A_C_* obtained by the Coast–Redfern method were compared with the activation energy *E_F_* obtained by the Flynn–Wall–Ozawa method and the pre-exponential factor *A_K_* obtained by the Kissinger method, respectively, and the correctness of the reaction mechanism function G(*α*) can be verified when the following conditions are satisfied simultaneously [43]: (15)$$\left|E_{F}-E_{C}\right|/E_{F}\leq 0.1,$$
(16)$$\left|\mathrm{lg}A_{C}-\mathrm{lg}A_{K}\right|/\mathrm{lg}A_{K}\leq 0.1,$$

By comparing the calculated values of several methods, the deviations are within 10%, which satisfies the requirements of Equations (15) and (16). It is proved that the above-estimated reaction mechanism function G(*α*) = ((1 − *α*)^−3.3^ − 1)/3.3 is applicable to the thermal decomposition reaction mechanism of the carbon fiber composite core.

### 3.5. Service Life Prediction

The aging process of carbon fiber composite cores is essentially a phenomenon of cracking and deterioration of the composite material, affected by factors such as light, temperature, and humidity, and the rate of chemical reaction will determine the service life of the material. Therefore, the remaining service life of the composite can be predicted by the chemical reaction rate and state parameters at a certain operating condition. 

That is, the service life *t* of a material in a certain operating state can be transformed to solve for a certain reaction mechanism function G(*α*) followed by material deterioration or cracking, and the reaction rate constant k(*T*) in that operating state, as shown in the following equation based on Equations (4) and (6):(17)$$t=\frac{G\left(\alpha \right)}{k\left(T\right)},$$

The end of life of the material is considered when the mass loss of the epoxy resin matrix composite reaches 5%, thus serving as a criterion for the end of life [44]. According to the TGA test results of the carbon fiber composite core, when its mass loss reaches 5%, the corresponding conversion rate α is about 0.34. The chemical reaction rate constant k(*T*) corresponding to different temperatures can be calculated from the pyrolysis kinetic parameters obtained from the study, and when substituted into Equation (17) together with the reaction mechanism function G(*α*), the service life of the carbon fiber composite core at different temperatures can be obtained, as shown in Figure 8.

As can be seen from Figure 8, the service life of the carbon fiber composite core gradually decreases as the working temperature of the conductor increases. According to the calculation results, when the continuous working temperature is 160 °C, the service life of the carbon fiber composite core is 20 years. When the operating temperature reaches 190 °C, near the glass transition temperature, the service life of the carbon fiber composite core is about 0.85 years. The typical continuous operating temperature of the conductor during its life cycle is 120–140 °C, with only short periods of operation above 160 °C for N-1 power system failure conditions. If the actual operating temperature of the carbon fiber composite core conductor is estimated, the continuous service time of the conductor should be more than 30 years, for example the life value at 150 °C (63 years).

The aging life prediction method used in the above study is meant to obtain the activation energy and reaction mechanism functions of the carbon fiber composite core through short-time experiments, and to predict the service life using the material’s intrinsic characteristic parameters. It should be noted that the life expectancy exhibited will vary due to the different types of base materials and curing agents used in different types of epoxy resins and the differences in curing processes. In theory, for the same thermal decomposition reaction, the thermodynamic parameters obtained by different methods should be basically the same within the error range and the mechanism function of the reaction should be uniquely determined, but in practice this is not the case. Due to the limitation of testing means, the kinetic study of thermal decomposition reactions is still macroscopic, and the conclusions obtained are only applicable to total package reactions. Since the “true” reaction mechanism function of solid thermal decomposition reactions is not known, the reaction mechanism function is often assumed to be a simple reaction, from which the apparent reaction level, activation energy, and pre-exponential factor are derived. However, solid thermal decomposition reactions are very complex, simple series often cannot describe the real kinetic processes of non-homogeneous solid reactions, and the actual processes may deviate from the assumed model.

In addition, considering the actual working environment of carbon fiber composite conductors, they will be affected by humidity, salt, acid, alkalis, ultraviolet radiation, and other environmental factors. Although the composite core of carbon fiber and epoxy resin is protected by an organic fiber layer and an aluminum wire layer outside, the above environmental factors may still have a certain impact on the aging of the material. Under the combined effect of external environmental factors and the carbon fiber composite core with its protective layer, the material in this state can be re-evaluated, and the remaining life is obtained by considering the degradation of key performance (such as mechanical tension, etc.) of the carbon fiber composite core conductor.

## 4. Conclusions

In this paper, a study was conducted on the service life of stranded carbon fiber composite core conductors for overhead transmission lines, and the following main conclusions and results were achieved:

(1) A study of the apparent activation energy of the carbon fiber composite core based on the TGA tests and thermal decomposition kinetics was carried out. According to the kinetic model of thermal decomposition of composite material, a variety of material activation energy calculation models were proposed, such as the Flynn–Wall–Ozawa, Kissinger, and Coast–Redfern methods. The activation energies of the stranded carbon fiber composite core were 168.76 kJ/mol, 166.79 kJ/mol, and 160.35 kJ/mol, respectively, based on the calculation models. The calculation results of these theories are basically close.

(2) The investigation of carbon fiber composite core life prediction method based on material pyrolysis kinetic theory and an activation energy calculation model was carried out. The prediction of the remaining service life of the composite material can be achieved by the reaction rate and state parameters in a certain operation state. The three kinetic state parameters of pyrolysis, namely activation energy, reaction mechanism function, and pre-exponential factor were obtained from the TGA curves to obtain the predicted service life of stranded carbon fiber composite cores. The service life of stranded carbon fiber composite core conductors was discussed based on the 5% thermal weight loss criterion of stranded carbon fiber composites. When the continuous working temperature is 160 °C, the service life of a carbon fiber composite core is 20 years. When the operating temperature reaches 190 °C, near the glass transition temperature, the service life of a carbon fiber composite core is about 0.85 years.

## Figures and Tables

**Figure 1 polymers-14-04431-f001:**
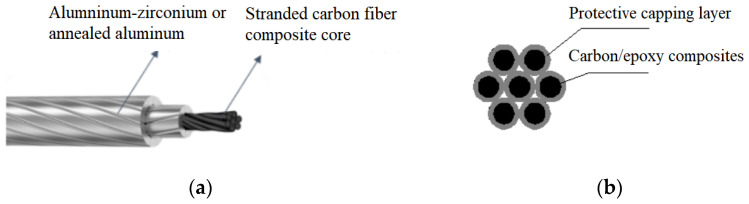
Structure of a stranded carbon fiber conductor: (**a**) carbon fiber composite core conductor; (**b**) stranded carbon fiber composite core.

**Figure 2 polymers-14-04431-f002:**
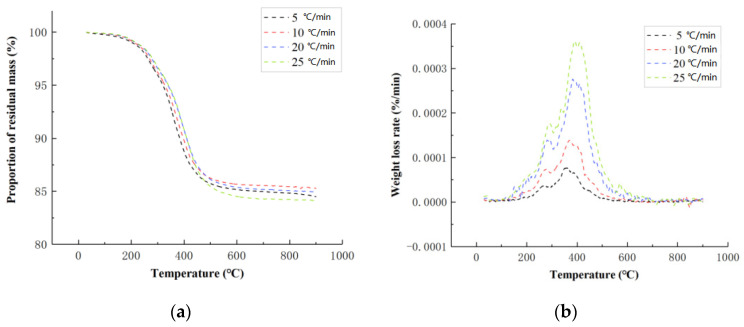
TGA test results of the carbon fiber composite core: (**a**) the thermogravimetric (TG) curves; (**b**) the derivative thermogravimetry (DTG) curves.

**Figure 3 polymers-14-04431-f003:**
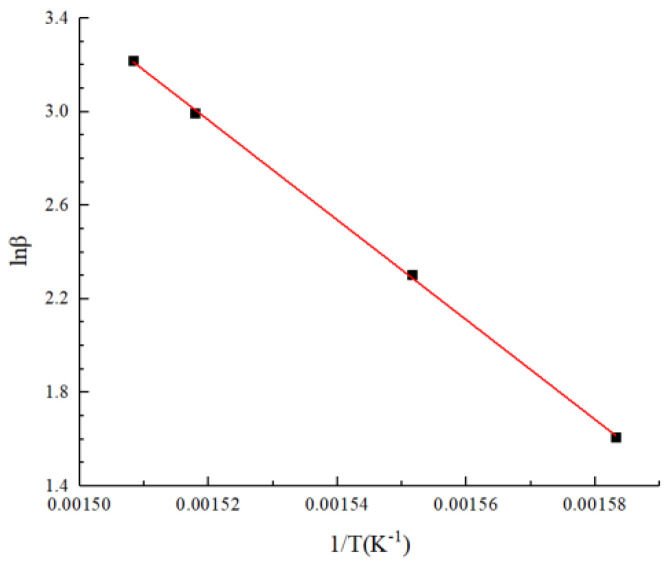
Linear fitting results at maximum thermal decomposition rate based on the Flynn–Wall–Ozawa method.

**Figure 4 polymers-14-04431-f004:**
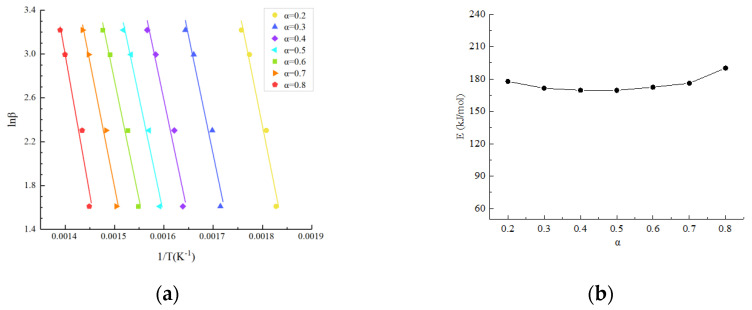
Results at different conversion rates based on the Flynn–Wall–Ozawa method: (**a**) linear fitting curves; (**b**) activation energy results.

**Figure 5 polymers-14-04431-f005:**
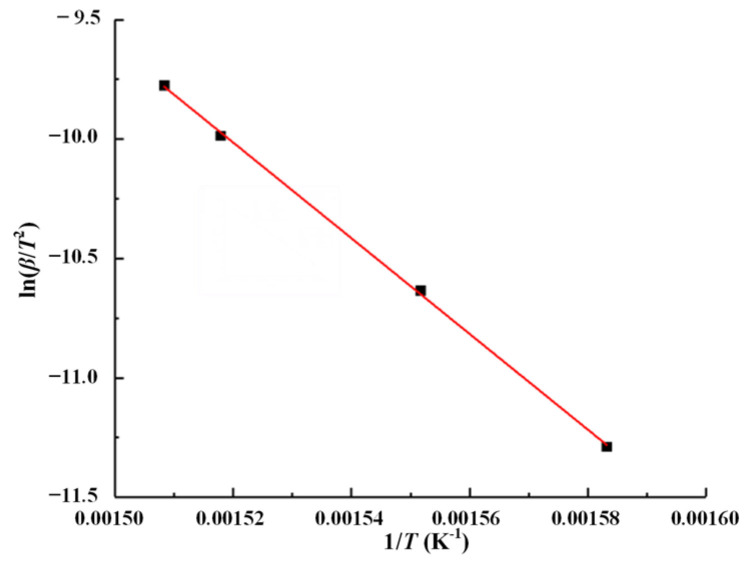
Linear fitting results at the maximum thermal decomposition rate based on the Kissinger method.

**Figure 6 polymers-14-04431-f006:**
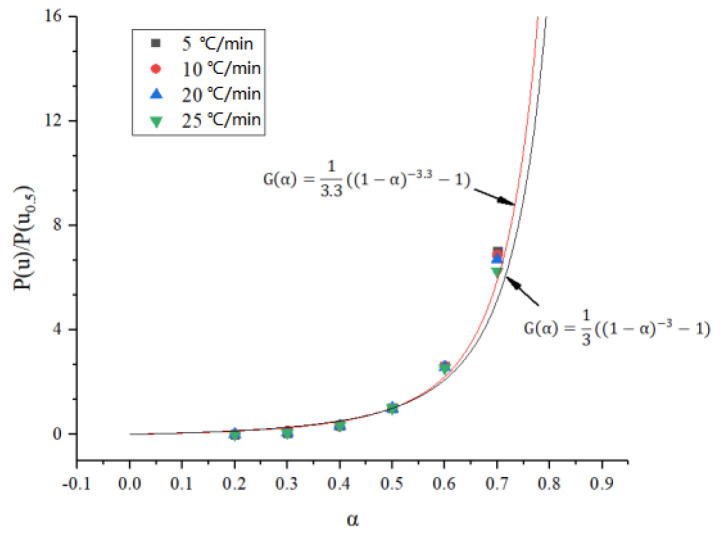
Fitting of the reaction mechanism function at different heating rates.

**Figure 7 polymers-14-04431-f007:**
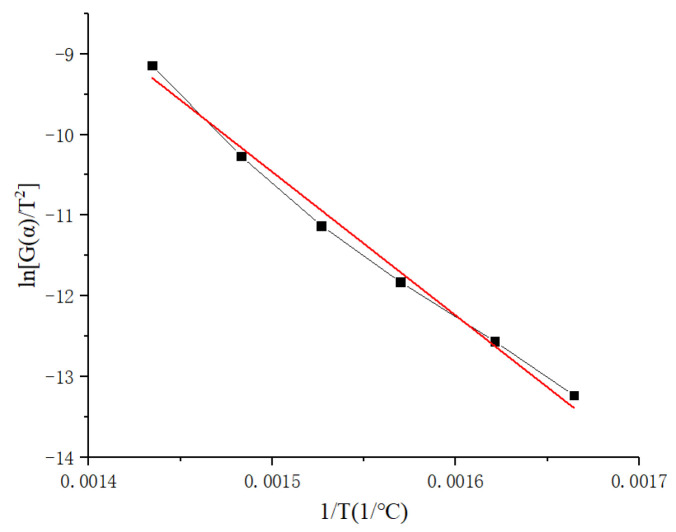
Linear fitting results based on the Coast–Redfern method.

**Figure 8 polymers-14-04431-f008:**
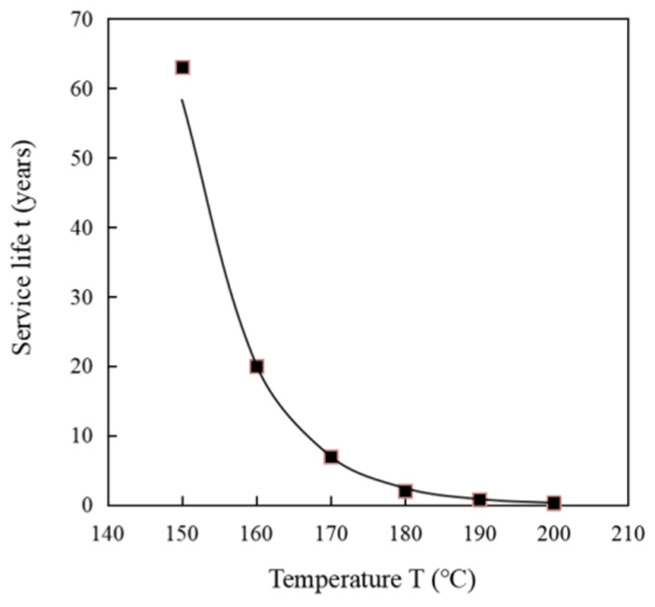
Service life at different temperatures.

**Table 1 polymers-14-04431-t001:** Peak temperature of DTG curve at different heating rates.

Heating Rate *β* (°C/min)	Peak Temperature *T_pi_* (°C)	Peak Temperature *T_pi_* (K)
5	359.500	631.650
10	372.333	644.483
20	386.667	658.817
25	390.833	662.983

## Data Availability

Data presented in this study are available on request from the first author.

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
