# Peer review of "A Service Life Prediction Method of Stranded Carbon Fiber Composite Core Conductor for Overhead Transmission Lines"

_polymers, 2022, doi:10.3390/polym14204431_

Round 1

Reviewer 1 Report

General Comments

The subject matter of the work is interesting and merits study. However, the presentation of the work leaves much to be desired. The abstract reads more like a paragraph from an introduction than an abstract. The abstract should be re-written using as few words as possible to answer these questions; what was done? How was it done? What are the most important findings/results from this work?

The introduction needs to be modified and include more relevant literature.

The assumptions on which the service life prediction was made are not justified.

Specific comments

LINE 10-13: This first sentence is better suited in the introduction section. Delete from the abstract.

LINE 13-16: Delete sentence.

LINE 16-22: Re-phrase this section in a succinct manner suitable for a scientific abstract. Authors might wish to read some abstracts in scientific literature published in leading journals in this area or from similar works.

LINE 27-28: Cite Figure 1 in this sentence so a reader can go to the illustration.

LINE 29: Delete “a”.

LINE 33: Delete “it”.

LINE 35: Write “capacity” instead of “the capacity”.

LINE 36-37: Make clear what is the “high temperature resistant conductor”. If it is “Fiber composite core conductor make this clear by inserting it in brackets before the end of the sentence.

LINE 39: Do not leave the reader guessing. Specifically indicate the “original conductor(s)” in brackets after mentioning “original conductor”.

LINE 41: For perspective include information on design life in a few other places in the world besides China and include citations.

LINE 58-60: Elaborate/highlight with citations the variety of factors besides humidity that contribute to carbon-fibre reinforced composite degradation especially when in contact with conductive metal and humidity/moisture as is bound to occur in humid weather conditions, before mentioning the method or factor(s) you considered in your work. See the following works for insights:

Ofoegbu, S. U., Ferreira, M. G., & Zheludkevich, M. L. (2019). Galvanically stimulated degradation of carbon-fiber reinforced polymer composites: a critical review. Materials, 12(4), p.651.  https://doi.org/10.3390/ma12040651

Santos, T. F. A., Vasconcelos, G. C., de Souza, W. A., Costa, M. L., & Botelho, E. C. (2015). Suitability of carbon fiber-reinforced polymers as power cable cores: Galvanic corrosion and thermal stability evaluation. Materials & Design (1980-2015), 65, 780-788. https://doi.org/10.1016/j.matdes.2014.10.005

Kotikalapudi, S. T., Akula, R., & Singh, R. P. (2022). Degradation mechanisms in carbon fiber–epoxy laminates subjected to constant low-density direct current. Composites Part B: Engineering, 233, p.109516. https://doi.org/10.1016/j.compositesb.2021.109516

Woo, E. M., Chen, J. S., & Carter, C. S. (1993). Mechanisms of degradation of polymer composites by galvanic reactions between metals and carbon fiber. Polymer composites, 14(5), pp.395-401. https://doi.org/10.1002/pc.750140505

LINE 63: “corrosion” or “degradation”?

LINE 68: “stress acceleration”? Please use the appropriate term.

LINE 66-70: Sentence is too long and lacks clarity. Rephrase using shorter sentences.

LINE 57-70: This paragraph is better suited for the “Material and Methods” section.

LINE 72-74: This sentence does not make sense; “The deterioration of materials actually also undergoes certain chemical reactions at different rates, which belong to pyrokinetic processes.” Please rephrase.

LINE 79: Write “evaluated” instead of “solved”.

LINE 80-83: Rephrase sentence to improve clarity. Use shorter sentences.

LINE 83: “aging” involves “deterioration”. Rephrase using either aging or deterioration but not both.

LINE 92: Insert references for this model/equation.

LINE 101: “mass of reaction” does not make sense.

LINE 118-121: Sentence is too long and lacks clarity. Rephrase using shorter sentences to improve clarity. Provide in brackets the producer and country of origin of the equipment.

LINE 124: Write “heating” instead of “warming”.

LINE 125: Incomplete sentence. “…….move toward” what?

LINE 131: Some explanation/insight  is needed on the temp peaks observed in Figure 2b.

LINE 169-173: This observation needs to be better discussed and compared with results from other workers to give some perspective.

LINE 232-233: Life prediction in this work is based on “”The end of life of the material is considered when the mass loss of the polymer  reaches 5%, thus serving as a criterion for the end of life.”.

However, this assumption is not shown to be based on any practical realities or published literature. Hence it can not be adjudged to be based on convincing facts to be indicative of degradation only.

LINE 307-362: A lot of the references do not have authors names; just initials. This is unacceptable.

Author Response

Dear reviewer:

Thank you for your decision and constructive comments on my manuscript. We have carefully considered the suggestion of reviewer and make some changes. We have tried our best to improve and made some changes in the manuscript.

The yellow part that has been revised according to your comments. Revision notes, point-to-point, are given as follows:

First of all, the abstract has been rewritten as required, focusing on what has been done in this article? How is it done? What are the most important findings/results of this work?

Secondly, the introduction is also modified according to the specific modification opinions, and the necessary references are cited.

Finally, the assumptions about life prediction are mainly derived from existing research results and cited source literature.

The following is the reply to the specific modification suggestions:

LINE 10-13: This first sentence is better suited in the introduction section. Delete from the abstract.

Reply: This part was modified according to the comment. Please see highlighted yellow section.

LINE 13-16: Delete sentence.

Reply: This part was modified according to the comment. Please see highlighted yellow section.

LINE 16-22: Re-phrase this section in a succinct manner suitable for a scientific abstract. Authors might wish to read some abstracts in scientific literature published in leading journals in this area or from similar works.

Reply: This part was modified according to the comment. Please see highlighted yellow section, line 10-24.

LINE 27-28: Cite Figure 1 in this sentence so a reader can go to the illustration.

Reply: It was modified according to the comment. Please see highlighted yellow section, line 30-31.

LINE 29: Delete “a”.

Reply: It was modified according to the comment. Please see highlighted yellow section, line 31.

LINE 33: Delete “it”.

Reply: It was modified according to the comment. Please see highlighted yellow section, line 35.

LINE 35: Write “capacity” instead of “the capacity”.

Reply: It was modified according to the comment. Please see highlighted yellow section, line 37.

LINE 36-37: Make clear what is the “high temperature resistant conductor”. If it is “Fiber composite core conductor make this clear by inserting it in brackets before the end of the sentence.

Reply: It was modified according to the comment. Please see highlighted yellow section, line 38-39.

LINE 39: Do not leave the reader guessing. Specifically indicate the “original conductor(s)” in brackets after mentioning “original conductor”.

Reply: It was modified according to the comment. Please see highlighted yellow section, line 41-42.

LINE 41: For perspective include information on design life in a few other places in the world besides China and include citations.

Reply: It was modified according to the comment. Please see highlighted yellow section, line 44. The following documents are referenced.

12.International Electrotechnical Commission. IEC 60826-2017 Overhead transmission lines-Design criteria, 2017.

13.Ministry of Housing and Urban Rural Integration of the People's Republic of China. GB 50545-2010 Code for design of 110kV~750kV overhead transmission line, 2010.

LINE 58-60: Elaborate/highlight with citations the variety of factors besides humidity that contribute to carbon-fibre reinforced composite degradation especially when in contact with conductive metal and humidity/moisture as is bound to occur in humid weather conditions, before mentioning the method or factor(s) you considered in your work. See the following works for insights:

Ofoegbu, S. U., Ferreira, M. G., & Zheludkevich, M. L. (2019). Galvanically stimulated degradation of carbon-fiber reinforced polymer composites: a critical review. Materials, 12(4), p.651.  https://doi.org/10.3390/ma12040651

Santos, T. F. A., Vasconcelos, G. C., de Souza, W. A., Costa, M. L., & Botelho, E. C. (2015). Suitability of carbon fiber-reinforced polymers as power cable cores: Galvanic corrosion and thermal stability evaluation. Materials & Design (1980-2015), 65, 780-788. https://doi.org/10.1016/j.matdes.2014.10.005

Kotikalapudi, S. T., Akula, R., & Singh, R. P. (2022). Degradation mechanisms in carbon fiber–epoxy laminates subjected to constant low-density direct current. Composites Part B: Engineering, 233, p.109516. https://doi.org/10.1016/j.compositesb.2021.109516

Woo, E. M., Chen, J. S., & Carter, C. S. (1993). Mechanisms of degradation of polymer composites by galvanic reactions between metals and carbon fiber. Polymer composites, 14(5), pp.395-401. https://doi.org/10.1002/pc.750140505

Reply: It was modified according to the comment. Please see highlighted yellow section, line 60-75. External pollution and moisture will affect the performance of carbon fiber composites. Therefore, the outer layer of twisted carbon fiber composites core is wrapped with organic fibers to prevent external corrosion. See the reference [25] for specific tests.

18.Ofoegbu, S. U.; Ferreira, M. G.; Zheludkevich M. L. Galvanically stimulated degradation of carbon-fiber reinforced polymer composites: a critical review. Materials, 2019, 12, 651.

19.Santos, T. F. A.; Vasconcelos, G. C.; de Souza; W. A.; Costa, M. L.; Botelho, E. C. Suitability of carbon fiber-reinforced polymers as power cable cores: Galvanic corrosion and thermal stability evaluation. Materials & Design, 2015, 65, 780-788. 

20.Kotikalapudi, S. T.; Akula, R.; Singh, R. P. Degradation mechanisms in carbon fiber–epoxy laminates subjected to constant low-density direct current. Composites Part B: Engineering, 2022, 233, 109516. 

21.Woo, E. M.; Chen, J. S.; Carter C. S. Mechanisms of degradation of polymer composites by galvanic reactions between metals and carbon fiber. Polymer composites, 1993, 14, 395-401. 

22.Xiaobo M.; Hongwei M.; Bo Z.; Fanghui Y.; Liming W. Influence of pollution on surface streamer discharge, Electric Power Systems Research, 2022, 212, 108638.

23.Sida Z.; Li C.; Ruijin L.; Yunfan L.; Xiying W.; Tingting W.; Junkai F. Process Improvement to Restrain Emergency Heating Defect of Composite Insulator, IEEE Transactions on Dielectrics and Electrical Insulation, 2022, 29, 446-453.

24.Li C.; Yunfan L.; Rongxin C.; Sida Z.; Ruijin L.; Lijun Y.; Tingting W. Method for predicting the water absorption of external insulating silicone rubber, IEEE Transactions on Dielectrics and Electrical Insulation, 2022, 29, 1242 - 1250.

25.Y. KONDO. High corrosion resistance of ACFR conductor. CIGRE-IEC 2019 Conference on EHV and UHV (AC & DC) . Hakodate, Japan, 23-26 April, 2019.

LINE 63: “corrosion” or “degradation”?

Reply: It was modified according to the comment. Please see highlighted yellow section, line 82.

LINE 68: “stress acceleration”? Please use the appropriate term.

Reply: It was modified according to the comment, “mechanical tensions”. Please see highlighted yellow section, line 60-75.

LINE 66-70: Sentence is too long and lacks clarity. Rephrase using shorter sentences.

Reply: It was modified according to the comment. Please see highlighted yellow section, line 86-92.

LINE 57-70: This paragraph is better suited for the “Material and Methods” section.

Reply: This paragraph was modified according to the comment. Please see highlighted yellow section, line 76-92. This article is mainly about the derivation and expression of the method. The logical idea is not written according to the article of experimental analysis, and it is not suggested to appear “Material and Methods” section separately.

LINE 72-74: This sentence does not make sense; “The deterioration of materials actually also undergoes certain chemical reactions at different rates, which belong to pyrokinetic processes.” Please rephrase.

Reply: I reread this sentence, which is wordy and repetitive and has been deleted. Please see highlighted yellow section, line 94-95.

LINE 79: Write “evaluated” instead of “solved”.

Reply: It was modified according to the comment. Please see highlighted yellow section, line 100.

LINE 80-83: Rephrase sentence to improve clarity. Use shorter sentences.

Reply: It was modified according to the comment. Please see highlighted yellow section, line 102-106.

LINE 83: “aging” involves “deterioration”. Rephrase using either aging or deterioration but not both.

Reply: It was modified according to the comment. Please see highlighted yellow section, line 104.

LINE 92: Insert references for this model/equation.

Reply: It was modified according to the comment. Please see highlighted yellow section, line 114. The references are as follows.

35.Shufen L.; Jiang Z.; Kaijun Y.; Shuqin Y. Studies on the Thermal Behavior of Polyurethanes. Polymer-Plastics Technology and Engineering. 2006, 45, 95-108.

LINE 101: “mass of reaction” does not make sense.

Reply: It was modified according to the comment. Please see highlighted yellow section, line 121-122. The statement has been revised according to relevant standards. The references are as follows.

  1. International Organization for Standardization. ISO 11358-2:2021 Plastics -- Thermogravimetry (TG) of polymers-- Part 2:Determination of activation energy, 2021.

LINE 118-121: Sentence is too long and lacks clarity. Rephrase using shorter sentences to improve clarity. Provide in brackets the producer and country of origin of the equipment.

Reply: It was modified according to the comment. Please see highlighted yellow section, line 141.

LINE 124: Write “heating” instead of “warming”.

Reply: It was modified according to the comment. Please see highlighted yellow section, line 147.

LINE 125: Incomplete sentence. “…….move toward” what?

Reply: It was modified according to the comment. Please see highlighted yellow section, line 150.

LINE 131: Some explanation/insight  is needed on the temp peaks observed in Figure 2b.

Reply: It was modified according to the comment. Please see highlighted yellow section, line 156-163.

LINE 169-173: This observation needs to be better discussed and compared with results from other workers to give some perspective.

Reply: This part was modified according to the comment. Please see highlighted yellow section, line 184-203

LINE 232-233: Life prediction in this work is based on “”The end of life of the material is considered when the mass loss of the polymer  reaches 5%, thus serving as a criterion for the end of life.”.

However, this assumption is not shown to be based on any practical realities or published literature. Hence it can not be adjudged to be based on convincing facts to be indicative of degradation only.

Reply: It was modified according to the comment. Please see highlighted yellow section, line 286-287. The references are as follows.

Zhang Z.; Liang G.; Ren P.; et al. Thermodegradation kinetics of epoxy/DDS/POSS system. Polymer Composites. 2010, 28, 755-761.

LINE 307-362: A lot of the references do not have authors names; just initials. This is unacceptable.

Reply: It was modified according to the comment. Please see highlighted yellow section, line 370-449.

Reviewer 2 Report

This manuscript reports a study of the service life of carbon fiber core conductor by TGA measurements. It is a useful research with certain merit for the field and it is recommended for publication except for a few minor problems in the methodology and analysis. The authors should address my comments before resubmission. 

1. In the introduction it is unclear to me how accurate and is the previous method and what are the main deficiencies of prior research on the service life of the carbon fiber composite. 

2. In section 3.1, the authors did not provide sufficient information about the exact type of the carbon fiber and the epoxy which makes it hard to interpret and compare the conclusions. Did the authors completely removed the metal shell of the cable before grinding and the TGA measurement? 

3. Please list all the activation energy and the prefactor A derived from Fig. 3 in a table in comparison with the result from Fig. 4(b). Besides, which curve from Fig. 3 is used in Fig. 4(a)? 

4. What value of activation energy E is used to calculate u in P(u) in the plot of the experimental data in Fig. 5? 

5. Is Eq. 13 consistent with Eq. (4)? A brief analysis to show the derivation is needed. 

6. A figure demonstration of the data in Table 1 will be helpful to visualize the trend of the service life variation with temperature.

Author Response

Dear reviewer:

Thank you for your decision and constructive comments on my manuscript. We have carefully considered the suggestion of reviewer and make some changes. We have tried our best to improve and made some changes in the manuscript.

The green part that has been revised according to your comments. Revision notes, point-to-point, are given as attached file.

  1. In the introduction it is unclear to me how accurate and is the previous method and what are the main deficiencies of prior research on the service life of the carbon fiber composite. 

Reply: In the introduction part, the research and method of aging test of composite materials for power equipment are described. The existing results show that the degradation of composite material performance is mainly obtained through accelerated aging test, and the conclusion about service life is given. The main purpose of this paper is to explore such a method and get some results.

  1. In section 3.1, the authors did not provide sufficient information about the exact type of the carbon fiber and the epoxy which makes it hard to interpret and compare the conclusions. Did the authors completely removed the metal shell of the cable before grinding and the TGA measurement? 

Reply: Sorry, the original statement is not clean enough. It has been rewritten, see the highlighted green section, line 136-140. The material used in the test is indeed to remove the outer metal aluminum of the transmission conductor and the outer organic fiber protective layer of the carbon fiber epoxy resin material. The test achieves the evaluation of carbon fiber epoxy composite system.

  1. Please list all the activation energy and the prefactor A derived from Fig. 3 in a table in comparison with the result from Fig. 4(b). Besides, which curve from Fig. 3 is used in Fig. 4(a)? 

Reply: Sorry that the original statement is not clear enough, which will lead to misunderstanding. The activation energies of different conversion rates in the range of 0.2-0.8 calculated by the Flynn-Wall-Ozawa method are expressed in graph form, but the factor A cannot be calculated by the Flynn-Wall-Ozawa method. The conversion rate corresponding to the fitting curve of Fig.4 (b) is different from the original Fig.3 Listed fit curves. It is fitted by the peak temperature corresponding to the maximum thermal decomposition rate in Fig.2 (b) of the DTG curve. It has been rewritten, see the highlighted green section, line 184-203.

  1. What value of activation energy E is used to calculate u in P(u) in the plot of the experimental data in Fig. 5? 

Reply: The activation energy data is derived from the newly supplemented diagram Fig. 4 (b). Please see the highlighted green section, line 204.

  1. Is Eq. 13 consistent with Eq. (4)? A brief analysis to show the derivation is needed. 

Reply: Eq. 13 is derived from the following references. Reaction mechanism function in the Eq. 13 is from simultaneous integration of both ends after term shifting with Eq. (4).

Coats A. W.; Redfern J. P. Kinetic parameters from thermogravimetric data. Nature. 1964, 201, 68-69.

  1. A figure demonstration of the data in Table 1 will be helpful to visualize the trend of the service life variation with temperature.

Reply: It has been changed to a graphic, which is more intuitive. Please see the highlighted green section, line 295.

Reviewer 3 Report

A manuscript entitled “A Service Life Prediction Method of Stranded Carbon Fiber Composite Core Conductor for Overhead Transmission Lines”  focuses on the TGA tests and thermal decomposition kinetics. Different calculation models have been proposed for estimation of energy activation. The overall impression of the manuscript is good. However, some important shortcomings were identified:

-          Why pyrolysis was used for life prediction of  carbon fiber composite core ? In line 60 Authors write:  “Differential scanning calorimetry and TGA methods were used to study the pyrolysis kinetics of composite insulators and to predict their service life [17].” However, reference 17 is not releted to this issue.

-          Subsection 3.5 Service Life Prediction: The model adopted by the Authors may make sense, but in relation to the required properties. The 5% reduction in weight itself may affect the usability of carbon fiber composite, but we do not know this. Perhaps 10% or 1% will result in deterioration of the required properties. The main task of conductors in transmission line is the electricity transport. The authors should answer the question of how the proposed research model based on TGA relates to the deterioration of the key properties of carbon fiber composites such as electrical conductivity.

Author Response

Dear reviewer:

Thank you for your decision and constructive comments on my manuscript. We have carefully considered the suggestion of reviewer and make some changes. We have tried our best to improve and made some changes in the manuscript.

Your comments, point-to-point, are given as follows:

Why pyrolysis was used for life prediction of  carbon fiber composite core ? In line 60 Authors write:  “Differential scanning calorimetry and TGA methods were used to study the pyrolysis kinetics of composite insulators and to predict their service life [17].” However, reference 17 is not releted to this issue.

Reply:The introduction of the article has been greatly revised. Why carbon fiber composites are used for thermal decomposition, and why other aging test methods for composites are referred to. We have noticed that in the previous aging studies on epoxy resin based composites, during the high-temperature decomposition process, the epoxy resin was first decomposed and carbonized, which led to the degradation of the properties of the composite system constructed. The research object of this article is the core body solidified by carbon fiber epoxy resin as the reinforcing core of the transmission conductor. In the transmission line, High current will pass through the transmission conductor, causing temperature rise. However, epoxy resin matrix composites are faced with the problem of decomposition at high temperatures. Therefore, the thermal aging of the transmission conductor core cured by carbon fiber epoxy resin is evaluated by referring to the existing research results on the thermal decomposition of epoxy resin based composites. The results of the reference 17 have been ignored in the revised edition.

Subsection 3.5 Service Life Prediction: The model adopted by the Authors may make sense, but in relation to the required properties. The 5% reduction in weight itself may affect the usability of carbon fiber composite, but we do not know this. Perhaps 10% or 1% will result in deterioration of the required properties. The main task of conductors in transmission line is the electricity transport. The authors should answer the question of how the proposed research model based on TGA relates to the deterioration of the key properties of carbon fiber composites such as electrical conductivity.

Reply:The criteria for the end of service life are mainly cited from the existing thermal aging test results of epoxy resin matrix composites. The references are as follows. However, the degradation of carbon fiber composites after thermal aging is indeed worthy of research and concern. Some comments and prospects were made in line 326-335. The main purpose of this article is to obtain the calculation formula of the life of carbon fiber composite through the derivation of the pyrolysis equation, and to scientifically explore the service life under thermal aging. I wonder if my reply is satisfactory. Thank you.

Zhang Z.; Liang G.; Ren P.; et al. Thermodegradation kinetics of epoxy/DDS/POSS system. Polymer Composites. 2010, 28, 755-761.

Reviewer 4 Report

In the manuscript “A Service Life Prediction Method of Stranded Carbon Fiber Composite Core Conductor for Overhead Transmission Lines”, the authors present a study to achieve effective assessment of stranded carbon fiber composite core conductor to avoid the degradation of mechanical properties and power accidents. From the comprehensive study of the apparent activation energy of carbon fiber composite core based on the TGA tests and thermal decomposition kinetics, the authors have demonstrated composite core life prediction theory. and I think the current manuscript is clearly written and can be published in Polymers.

Author Response

Dear reviewer:

Thank you for your decision and constructive comments on my manuscript. We have carefully considered the suggestion of other reviewer and make some changes. We have tried our best to improve and made some changes in the manuscript. Please review again. 

Reviewer 5 Report

OVERVIEW

The authors deal with the issue related to the aging degree and remaining life of stranded carbon fiber composite core conductor of overhead transmission lines. The authors performed thermal decomposition activation energy calculation of the carbon fiber composite core by thermogravimetric analysis. The authors performed heat loss tests at five different temperature ramping rates of 5, 10, 20, and 25 K/min on carbon fiber composite samples to obtain the heat loss curve variation law. The authors found out that the thermal weight loss process of the carbon fiber composite core follows a single reaction mechanism function. The authors determined the remaining lifetime of carbon fiber composites at various temperatures by calculating the activation energy, reaction mechanism function, and pre-exponential factor.

POSITIVE ASPECTS

1. Based on a literature review, the authors made an overview of the issues related to the aging process of stranded carbon fiber composite core conductor.
2. The authors claim that the physical mechanisms and mechanisms of the occurrence and development of thermal aging of carbon fiber composites are not fully understood.
3. Based on a literature review, the authors reviewed the methodologies used to experimentally determine the approximate operating life of carbon fiber composite conductor.
4. The authors found out that the thermal weight loss process of the carbon fiber composite core follows a single reaction mechanism function.
5. The authors obtained three kinetic state parameters of pyrolysis to determine the remaining lifetime of carbon fiber composites.

ISSUES

The presented work is useful but has some issues that need to be removed. I have a few comments that can be used to improve the article.

Minor issues
1. The authors use a dual way of marking physical quantities – compare, for example, the marking in Equation 1 on page 3, and lines 94, 95; Equation 2, and in lines 100 and 101. Correct it, and use a unified way of marking physical quantities throughout the text, including descriptions of the figures and axis designations in the graphs.
2. Signs of physical quantities need to write in italics according to ISO 31-4 (ISO 80000-5: 2007). Corrections need to be done throughout the article, including tables and figures.
3. The sign for multiplication of letter symbol to a half-high dot (·) is preferred; see line 105.
4. The differential “d” of some variable is always in roman (upright) type, not in italics (use “dx”, not “dx”); see, for example, line 108. Correct it throughout the text.
5. The numerical value of the physical unit and its symbol shall be separated by a space according to ISO 80000-1: 2009 standard (e.g. page 4, lines 118, 121). Correct all text accordingly, including tables and figures.
6. The authors use a dual way of naming some terms. Compare the term “integral form” and the term “Integral form.” Correct accordingly throughout the text.
7. The mathematical symbols (functions) should write in upright type; see lines 143 to 144. Correct accordingly throughout the text.
8. Do not use a hyphen (-) instead of a minus sign (–). Correct accordingly throughout the text.
9. The authors use a dual way of writing the mark to express the reaction conversion rate (Greek letter alpha). Correct all text accordingly.
10. Arabic numerals are written in upright type, for example lines 182 and 187. This also applies to indices in equations and the marks of physical quantities and superscripts. Correct accordingly throughout the text.
11. The parentheses are written in a straight font; see line 183. Correct accordingly throughout the text.

Major issues
1. The authors claim in lines 232 to 233 that the end of life of the material is considered when the mass loss of the polymer reaches 5%, thus serving as a criterion for the end of life. A corresponding reference to the literature should be added to this statement.

RECOMMENDATIONS

1. It would be more appropriate to supplement the article with the Materials and methods section.
2. I recommend the authors reformulate the paragraph in lines 268 to 274 so that the intention of future experiments regarding the service life of stranded carbon fiber composite core conductor for overhead transmission lines emerges from it.
3. In the Conclusion section, I recommend the authors include a text summarizing the results from Table 1.

QUESTIONS

I have one question for the authors of the article.

1. Why do the authors comment on cross-linked epoxy resin molecules in lines 171 to 173 when in lines 115 to 117 they state that the organic fiber filaments wound on the surface of the unused stranded carbon fiber composite core were filed off? Is this a misunderstanding on my part? If not, add a corresponding comment to the text.

CONCLUSION

I find this article helpful. Regretfully, the paper cannot be accepted in its present form. The authors of the present article have to correct the issues.

Author Response

Dear reviewer:

Thank you for your decision and constructive comments on my manuscript. We have carefully considered the suggestion of reviewer and make some changes. We have tried our best to improve and made some changes in the manuscript.

The pink part that has been revised according to your comments. Revision notes, point-to-point, are given as follows:

Minor issues

  1. The authors use a dual way of marking physical quantities – compare, for example, the marking in Equation 1 on page 3, and lines 94, 95; Equation 2, and in lines 100 and 101. Correct it, and use a unified way of marking physical quantities throughout the text, including descriptions of the figures and axis designations in the graphs.
  2. Signs of physical quantities need to write in italics according to ISO 31-4 (ISO 80000-5: 2007). Corrections need to be done throughout the article, including tables and figures.
  3. The sign for multiplication of letter symbol to a half-high dot (·) is preferred; see line 105.
  4. The differential “d” of some variable is always in roman (upright) type, not in italics (use “dx”, not “dx”); see, for example, line 108. Correct it throughout the text.
  5. The numerical value of the physical unit and its symbol shall be separated by a space according to ISO 80000-1: 2009 standard (e.g. page 4, lines 118, 121). Correct all text accordingly, including tables and figures.
  6. The authors use a dual way of naming some terms. Compare the term “integral form” and the term “Integral form.” Correct accordingly throughout the text.
  7. The mathematical symbols (functions) should write in upright type; see lines 143 to 144. Correct accordingly throughout the text.
  8. Do not use a hyphen (-) instead of a minus sign (–). Correct accordingly throughout the text.
  9. The authors use a dual way of writing the mark to express the reaction conversion rate (Greek letter alpha). Correct all text accordingly.
  10. Arabic numerals are written in upright type, for example lines 182 and 187. This also applies to indices in equations and the marks of physical quantities and superscripts. Correct accordingly throughout the text.
  11. The parentheses are written in a straight font; see line 183. Correct accordingly throughout the text.

Reply: Thank you for your suggestions on standard format requirements for mathematical functions. I have revised the full text in accordance with this opinion. Please review again.

Major issues

  1. The authors claim in lines 232 to 233 that the end of life of the material is considered when the mass loss of the polymer reaches 5%, thus serving as a criterion for the end of life. A corresponding reference to the literature should be added to this statement.

Reply: The criteria for the end of service life are mainly cited from the existing thermal aging test results of epoxy resin matrix composites. The references are as follows.

Zhang Z.; Liang G.; Ren P.; et al. Thermodegradation kinetics of epoxy/DDS/POSS system. Polymer Composites. 2010, 28, 755-761.

RECOMMENDATIONS

  1. It would be more appropriate to supplement the article with the Materials and methods section.

Reply: This article is mainly about the derivation and expression of the method. The logical idea is not written according to the article of experimental analysis, and it is not suggested to appear “Material and Methods” section separately.

  1. I recommend the authors reformulate the paragraph in lines 268 to 274 so that the intention of future experiments regarding the service life of stranded carbon fiber composite core conductor for overhead transmission lines emerges from it.

Reply: Some comments and prospects were made in line 326-335. The main purpose of this article is to obtain the calculation formula of the life of carbon fiber composite through the derivation of the pyrolysis equation, and to scientifically explore the service life under thermal aging. I wonder if my reply is satisfactory. Thank you.

  1. In the Conclusion section, I recommend the authors include a text summarizing the results from Table 1.

Reply: In the conclusion part, the numerical results of the main results are supplemented. Please see line 354-359.

QUESTIONS

I have one question for the authors of the article.

  1. Why do the authors comment on cross-linked epoxy resin molecules in lines 171 to 173 when in lines 115 to 117 they state that the organic fiber filaments wound on the surface of the unused stranded carbon fiber composite core were filed off? Is this a misunderstanding on my part? If not, add a corresponding comment to the text.

Reply: That's true. The object of this paper is the composite cured by carbon fiber and epoxy resin. The actual carbon fiber composite core is composed of carbon fiber epoxy resin material and a protective layer covered with organic fibers. The main purpose of the protective layer is to protect the carbon fiber epoxy resin material, prevent the galvanic corrosion of different material interfaces, as well as the corrosion of dirt and moisture. The description of the test object has been modified, please see line 136-140.

Round 2

Reviewer 1 Report

REVIEWER COMMENTS__ROUND 2__ A Service Life Prediction Method of Stranded Carbon Fiber Composite Core Conductor for Overhead Transmission Lines.

General  Comments

The clarity of the manuscript has been significantly improved. Important materials and equations omitted in the earlier version have been added. However, the abstract still needs to be improved.

The claim of no galvanic degradation in aluminium carbon fiber composite couples need to be modified as several reports report this occurs but is smaller than steel- carbon fiber composite couples.

A few references are missing to support some statements made by authors.

Suggestions to rectify a few errors which would improve the quality of the manuscript has been provided under specific comments

Specific Comments

LINE 10-11: Write “The effect of temperature on the service life of stranded carbon fiber composite core conductor was studied based on the kinetic theory of material pyrolysis.” instead of “A study of the effect of temperature on the service life of stranded carbon fiber composite 10core conductor based on the kinetic theory of material pyrolysis.”

LINE 15: Write “The results from these different treatments were within” instead of “The mutual results were within”.

LINE 17: Delete “Then,”.

LINE 21-24: Delete the last 2 sentences. Add a single sentence that tells the reader the most important finding from your work (The key take-away from your work).

LINE 35: Write “smaller” instead of “small”.

LINE 60: Put a full-stop after “complex”. Delete “, in”.

LINE 61: Start a new sentence with “The atmospheric…”.

LINE 70-72: “Stranded carbon fiber composite core conductor has a protective layer of the composite core effectively avoids the galvanic coupling corrosion of the steel core aluminum conductor.”

This statement is not consistent with several literature reports of galvanic corrosion between carbon fiber composites and aluminium. See the following literature. Acknowledge this occurrence or possibility and present your argument in a different way that any galvanic corrosion effects will be less severe compared to steel-aluminum couple.

Wu, X., Sun, J., Wang, J., Jiang, Y., & Li, J. (2019). Investigation on galvanic corrosion behaviors of CFRPs and aluminum alloys systems for automotive applications. Materials and Corrosion, 70(6), 1036-1043.   https://doi.org/10.1002/maco.201810635#

Ofoegbu, S. U., Ferreira, M. G., & Zheludkevich, M. L. (2019). Galvanically stimulated degradation of carbon-fiber reinforced polymer composites: a critical review. Materials, 12(4), 651. https://doi.org/10.3390/ma12040651

Santos, T. F. A., Vasconcelos, G. C., de Souza, W. A., Costa, M. L., & Botelho, E. C. (2015). Suitability of carbon fiber-reinforced polymers as power cable cores: Galvanic corrosion and thermal stability evaluation. Materials & Design (1980-2015), 65, 780-788. https://doi.org/10.1016/j.matdes.2014.10.005

LINE 78-79: “Many scholars have used various methods to study the service life of com-78posite materials for power facilities”  Citations of these earlier works should be inserted at the end of this sentence.

LINE 79: Write “Fickian” instead of “Fick-ian”

LINE 87: Write “stresses” instead of “tensions”.

LINE 88: Write “stresses” instead of “tensions”.

LINE 90: Write “stresses” instead of “tensions”.

LINE 100: Write “cause” instead of “effect”.

LINE 106: Delete “law”.

LINE 110: Write “evaluated” instead of “given”.

LINE 120: Write “mass” instead of “quantity”.

LINE 121: Write “mass” instead of “quantity”. (2 cases).

LINE 128: Write “for” instead of “of”.

LINE 131 Write “can” instead of “will”.

LINE 138-139: Write “it is ground into powder and placed in drying oven at 40 °C for drying” instead of “grind it into powder and place it in a constant temperature drying oven at 40 °C for drying”.

LINE 141: Write “was carried out in a dry….” Instead of “was in a dry…..”.

LINE 141-142: Delete “with quantity”.

LINE 141: Write “Samples” instead of “The sample”.

LINE 148: Write from thermogravimetric data” instead of “of thermogravimetry”.

LINE 148: Write “increase” instead of “the increase”.

LINE 149: Write “peaks in both TG and DTG curves” instead of “both the TG and DTG curves”.

LINE 154: Delete “functions”.

LINE 196: “thermal weightlessness test”??  This term is vague and very confusing. What do you mean by this? Rephrase sentences using acceptable terms to improve clarity.

LINE 197: Write “is predominantly due to the decomposition..” instead of “is mainly the decomposition….”.

LINE 199: Delete “to participate”.

LINE 210: Write “is equal to -E/R”

LINE 213: Write “and” instead of “And”.

LINE 233: Write “analytical”

LINE 279: Write “to solve” instead of “ into solving”.

LINE 283: Delete “mates”.

Author Response

Dear reviewer:

Thank you very much for your valuable comments and constructive suggestions with regard to our manuscript. These comments are helpful for us to revise and improve our paper. We have studied the comments carefully and tried our best to revise and improve the manuscript and made great changes in the manuscript according to the comments. The revised part is marked in red on the paper. We appreciate for Reviewer’s warm work earnestly and hope that the corrections will meet with approval. Please feel free to contact us with any questions and we are looking forward to your consideration. The main corrections in the paper and the responses to the reviewer’ comments are as follows:

General  Comments

The clarity of the manuscript has been significantly improved. Important materials and equations omitted in the earlier version have been added. However, the abstract still needs to be improved.

Reply: Thank you for your comments. The abstract part has been revised according to specific suggestions, highlighting the main findings and results of this paper. 

The claim of no galvanic degradation in aluminium carbon fiber composite couples need to be modified as several reports report this occurs but is smaller than steel- carbon fiber composite couples.

Reply: Thank you for your comments. For the problem of galvanic corrosion, the corresponding writing method is modified. The existing research results are enough to support the conclusion that there is no galvanic corrosion, which can only be said to be mitigated. Compared with ACSR, it is more corrosion resistant. It has been corrected to “Stranded carbon fiber composite core conductor has a protective layer surrounding the composite core which makes the galvanic corrosion less severe compared to the steel-core aluminum stranded conductor[26].” Line 67-69.

A few references are missing to support some statements made by authors.

Reply: Thank you for your comments. The references have been added according to the comments.

Suggestions to rectify a few errors which would improve the quality of the manuscript has been provided under specific comments

Reply: Thank you very much for your valuable comments. We have corrected the manuscript point-by-point according to the comments.

The following is the reply to the specific modification suggestions:

LINE 10-11: Write “The effect of temperature on the service life of stranded carbon fiber composite core conductor was studied based on the kinetic theory of material pyrolysis.” instead of “A study of the effect of temperature on the service life of stranded carbon fiber composite core conductor based on the kinetic theory of material pyrolysis.”

Reply: This part was modified according to the comment. Please see highlighted red section. Line 10-11.

LINE 15: Write “The results from these different treatments were within” instead of “The mutual results were within”.

Reply: This part was modified according to the comment. Please see highlighted red section. Line 16.

LINE 17: Delete “Then,”.

Reply: This part was modified according to the comment. Please see highlighted red section. Line 17.

LINE 21-24: Delete the last 2 sentences. Add a single sentence that tells the reader the most important finding from your work (The key take-away from your work).

Reply: This part was modified according to the comment. Please see highlighted red section. The last sentence describes the main findings and results of this paper. Line 22-24.

LINE 35: Write “smaller” instead of “small”.

Reply: This part was modified according to the comment. Please see highlighted red section. Line 35.

LINE 60: Put a full-stop after “complex”. Delete “, in”.

Reply: This part was modified according to the comment. Please see highlighted red section. Line 60.

LINE 61: Start a new sentence with “The atmospheric…”.

Reply: This part was modified according to the comment. Please see highlighted red section. Line 61.

LINE 70-72: “Stranded carbon fiber composite core conductor has a protective layer of the composite core effectively avoids the galvanic coupling corrosion of the steel core aluminum conductor.”

This statement is not consistent with several literature reports of galvanic corrosion between carbon fiber composites and aluminium. See the following literature. Acknowledge this occurrence or possibility and present your argument in a different way that any galvanic corrosion effects will be less severe compared to steel-aluminum couple.

Wu, X., Sun, J., Wang, J., Jiang, Y., & Li, J. (2019). Investigation on galvanic corrosion behaviors of CFRPs and aluminum alloys systems for automotive applications. Materials and Corrosion, 70(6), 1036-1043.   https://doi.org/10.1002/maco.201810635#

Ofoegbu, S. U., Ferreira, M. G., & Zheludkevich, M. L. (2019). Galvanically stimulated degradation of carbon-fiber reinforced polymer composites: a critical review. Materials, 12(4), 651. https://doi.org/10.3390/ma12040651

Santos, T. F. A., Vasconcelos, G. C., de Souza, W. A., Costa, M. L., & Botelho, E. C. (2015). Suitability of carbon fiber-reinforced polymers as power cable cores: Galvanic corrosion and thermal stability evaluation. Materials & Design (1980-2015), 65, 780-788. https://doi.org/10.1016/j.matdes.2014.10.005

Reply: I agree with the reviewers. The existing research results are enough to support the conclusion that there is no galvanic corrosion, which can only be said to be mitigated. Compared with ACSR, it is more corrosion resistant. This part was modified according to the comment. Please see highlighted red section. Line 63-69.

LINE 78-79: “Many scholars have used various methods to study the service life of com-78posite materials for power facilities”  Citations of these earlier works should be inserted at the end of this sentence.

Reply: This part was modified according to the comment. Please see highlighted red section. Line 73.

LINE 79: Write “Fickian” instead of “Fick-ian”

Reply: This part was modified according to the comment. Please see highlighted red section. Line 73.

LINE 87: Write “stresses” instead of “tensions”.

Reply: This part was modified according to the comment. Please see highlighted red section. Line 82.

LINE 88: Write “stresses” instead of “tensions”.

Reply: This part was modified according to the comment. Please see highlighted red section. Line 83.

LINE 90: Write “stresses” instead of “tensions”.

Reply: This part was modified according to the comment. Please see highlighted red section. Line 84.

LINE 100: Write “cause” instead of “effect”.

Reply: This part was modified according to the comment. Please see highlighted red section. Line 95.

LINE 106: Delete “law”.

Reply: This part was modified according to the comment. Please see highlighted red section. Line 100.

LINE 110: Write “evaluated” instead of “given”.

Reply: This part was modified according to the comment. Please see highlighted red section. Line 104.

LINE 120: Write “mass” instead of “quantity”.

Reply: This part was modified according to the comment. Please see highlighted red section. Line 114-115.

LINE 121: Write “mass” instead of “quantity”. (2 cases).

Reply: This part was modified according to the comment. Please see highlighted red section. Line 114-115.

LINE 128: Write “for” instead of “of”.

Reply: This part was modified according to the comment. Please see highlighted red section. Line 118.

LINE 131 Write “can” instead of “will”.

Reply: This part was modified according to the comment. Please see highlighted red section. Line 125.

LINE 138-139: Write “it is ground into powder and placed in drying oven at 40 °C for drying” instead of “grind it into powder and place it in a constant temperature drying oven at 40 °C for drying”.

Reply: This part was modified according to the comment. Please see highlighted red section. Line 132-133.

LINE 141: Write “was carried out in a dry….” Instead of “was in a dry…..”.

Reply: This part was modified according to the comment. Please see highlighted red section. Line 135.

LINE 141-142: Delete “with quantity”.

Reply: This part was modified according to the comment. Please see highlighted red section. Line 136.

LINE 141: Write “Samples” instead of “The sample”.

Reply: This part was modified according to the comment. Please see highlighted red section. Line 136.

LINE 148: Write from thermogravimetric data” instead of “of thermogravimetry”.

Reply: This part was modified according to the comment. Please see highlighted red section. Line 142.

LINE 148: Write “increase” instead of “the increase”.

Reply: This part was modified according to the comment. Please see highlighted red section. Line 143.

LINE 149: Write “peaks in both TG and DTG curves” instead of “both the TG and DTG curves”.

Reply: This part was modified according to the comment. Please see highlighted red section. Line 143.

LINE 154: Delete “functions”.

Reply: This part was modified according to the comment. Please see highlighted red section. Line 149.

LINE 196: “thermal weightlessness test”??  This term is vague and very confusing. What do you mean by this? Rephrase sentences using acceptable terms to improve clarity.

Reply: Sorry for the mistakes. It should be the thermogravimetric analysis (TGA) and it is updated in the revised manuscript. This part was modified according to the comment. Please see highlighted red section. Line 190-191.

LINE 197: Write “is predominantly due to the decomposition..” instead of “is mainly the decomposition….”

Reply: This part was modified according to the comment. Please see highlighted red section. Line 192.

LINE 199: Delete “to participate”.

Reply: This part was modified according to the comment. Please see highlighted red section. Line 194.

LINE 210: Write “is equal to -E/R”

Reply: This part was modified according to the comment. Please see highlighted red section. Line 205.

LINE 213: Write “and” instead of “And”.

Reply: This part was modified according to the comment. Please see highlighted red section. Line 209.

LINE 233: Write “analytical”

Reply: This part was modified according to the comment. Please see highlighted red section. Line 230.

LINE 279: Write “to solve” instead of “ into solving”.

Reply: This part was modified according to the comment. Please see highlighted red section. Line 277.

LINE 283: Delete “mates”.

Reply: This part was modified according to the comment. Please see highlighted red section. Line 281.